# Neutralization of SARS-CoV-2 variants by convalescent and BNT162b2 vaccinated serum

Timothy A. Bates [1,8], Hans C. Leier [1,8], Zoe L. Lyski [1], Savannah K. McBride[1], Felicity J. Coulter [1], Jules B. Weinstein [1], James R. Goodman[2], Zhengchun Lu[1], Sarah A. R. Siegel [3], Peter Sullivan[3], Matt Strnad[3], Amanda E. Brunton[3], David X. Lee[1], Andrew C. Adey [4,5], Benjamin N. Bimber [6], Brian J. O'Roak[4], Marcel E. Curlin [7✉], William B. Messer [1,3,7✉] & Fikadu G. Tafesse [1✉]

SARS-CoV-2 and its variants continue to infect hundreds of thousands every day despite the rollout of effective vaccines. Therefore, it is essential to understand the levels of protection that these vaccines provide in the face of emerging variants. Here, we report two demographically balanced cohorts of BNT162b2 vaccine recipients and COVID-19 patients, from which we evaluate neutralizing antibody titers against SARS-CoV-2 as well as the B.1.1.7 (alpha) and B.1.351 (beta) variants. We show that both B.1.1.7 and B.1.351 are less well neutralized by serum from vaccinated individuals, and that B.1.351, but not B.1.1.7, is less well neutralized by convalescent serum. We also find that the levels of variant-specific anti-spike antibodies are proportional to neutralizing activities. Together, our results demonstrate the escape of the emerging SARS-CoV-2 variants from neutralization by serum antibodies, which may lead to reduced protection from re-infection or increased risk of vaccine breakthrough.

[1] Department of Molecular Microbiology & Immunology, Oregon Health & Science University (OHSU), Portland, OR, USA. [2] Medical Scientist Training Program, OHSU, Portland, OR, USA. [3] OHSU-PSU School of Public Health, Program in Epidemiology, Portland, OR, USA. [4] Department of Molecular & Medical Genetics, OHSU, Portland, OR, USA. [5] Knight Cardiovascular Institute, OHSU, Portland, OR, USA. [6] Vaccine and Gene Therapy Institute, OHSU, Beaverton, OR, USA. [7] Department of Medicine, Division of Infectious Diseases, OHSU, Portland, OR, USA. [8] These authors contributed equally: Timothy A. Bates, Hans C. Leier. ✉email: curlin@ohsu.edu; messer@ohsu.edu; tafesse@ohsu.edu

Since its emergence in Wuhan, China in late 2019, severe acute respiratory syndrome coronavirus 2 (SARS-CoV-2) has spread worldwide, causing widespread illness and mortality from coronavirus 2019 disease (COVID-19)[1]. Continued SARS-CoV-2 transmission has led to the emergence of variants of concern (VOCs) that show evidence of increased transmissibility or resistance to prior immunity[2,3]. By early 2021, three major VOCs were widely recognized: B.1.1.7, also called variant alpha;[4] B.1.351, also called variant beta; and P.1, also called variant gamma[4,5]. These VOCs were associated with increases in infections and hospitalizations in their countries of origin, and all have increased in frequency in other regions, suggesting a competitive fitness advantage over existing lineages[6].

Though a relatively small number of nonsynonymous mutations and deletions distinguish VOCs from earlier lineages (Supplementary Table 1), many of these encode residues in the spike protein, which interacts with the SARS-CoV-2 cellular receptor, angiotensin-converting enzyme 2 (ACE2), via its receptor-binding domain (RBD)[7,8]. RBD mutations could potentially increase transmissibility by enhancing binding to ACE2, or promote immune escape by altering epitopes that are the primary target of potently neutralizing antibodies[8]. In fact, the most prominent mutation that appeared early in the pandemic and rose to near-fixation in new strains was a substitution at spike residue position 614 (D614G) which positions the RBD in a more accessible configuration and confers greater infectivity but also greater susceptibility to neutralizing antibodies[9,10].

In addition to sharing D614G and a N501Y substitution which is associated with greater ACE2 affinity[11], VOCs have acquired other spike mutations, some of which are associated with resistance to antibody neutralization. These include E484K and K417N/T, both of which arose independently in the B.1.351 and P.1 lineages[12–14]. Epidemiological reports suggest that natural immunity to earlier SARS-CoV-2 lineages may confer limited protection from reinfection by B.1.351 or P.1[4,15], and prior analyses using relatively small numbers of vaccinee sera against pseudotyped or chimeric viruses showed reduced neutralization of B.1.351 and P.1[14,16].

In this study, we use clinical virus isolates of SARS-CoV-2, the B.1.1.7 variant, and the B.1.351 variant to examine the potency of the antibody response to both BNT162b2 vaccination and natural infection. We find that vaccinated serum is less effective at neutralizing B.1.1.7 and B.1.351 than early lineage SARS-CoV-2, and that convalescent serum is less neutralizing against B.1.351, but B.1.1.7 is similarly neutralized compared to early lineage SARS-CoV-2. Further, we find that age correlates negatively with vaccine response against both variants, and that following natural infection, neutralizing antibody titers wane to undetectable levels within 6 months to a year after infection.

## Results

**Antibody response to BNT162b2 vaccination.** The three COVID-19 vaccines authorized for emergency use by the U.S. Food and Drug Administration (BNT162b2 [Pfizer–BioNTech], mRNA-1273 [Moderna], and Ad26.COV2.S [Janssen]) elicit immunity using a spike protein antigen derived from early isolates such as USA_WA1/2020 (WA1)[17]. Numerous reports have shown that the mutant spike proteins expressed by the VOCs may bind less strongly to antibody repertoires induced by these vaccines[18–20]. RBD-binding antibody levels in adults who had received two doses of the BNT162b2 mRNA vaccine were determined by ELISA using recombinant RBD from WA1 (RBD-WA1) and RBDs with all amino acid substitutions possessed by each B.1.1.7 (N501Y) and B.1.351 (N501Y, E484K, K417N) (Supplementary Table 1). Compared to that of RBD-WA1,

vaccinated patient sera had a geometric mean 50% effective concentrations (EC50) which were 1.4-fold lower ($P = 0.0089$) for RBD-B.1.1.7 and 1.5-fold lower ($P = 0.0351$) for RBD-B.1.351 (Fig. 1a). BNT162b2-elicited antibodies also displayed potent neutralizing activity against WA1 in a 50% focus reduction neutralization test (FRNT50) (geometric mean titer (GMT) 1:393 ± 2.5) but decreased neutralization of B.1.1.7 (GMT 1:149 ± 2.4) and B.1.351 (GMT 1:45 ± 2.3), representing 2.6-fold ($P < 0.0001$) and 8.8-fold ($P < 0.0001$) reductions, respectively (Fig. 1b and Supplementary Fig. 1). The positive correlation between serum EC50 and FRNT50 was consistent for each matched variant-RBD pair, indicating that variant-specific RBD-targeted antibody concentration is proportional to live virus neutralization capacity against each lineage (Fig. 1c).

We next compared the FRNT50 values of each patient for WA1 to their FRNT50 values for each of the variants, finding that neutralizing titers for WA1 and B.1.1.7 were highly correlated at the individual level (Fig. 1d). In contrast, WA1 and B.1.351 FRNT50 titers correlated weakly at the individual level, with some individuals' sera able to potently neutralize WA1 while simultaneously failing to neutralize B.1.351 at the highest concentration used in our assay (1:20) (Fig. 1e). The lower correlation between B.1.351 with WA1 FRNT50 values likely indicates that a larger proportion of the epitopes recognized by WA1 neutralizing antibodies are functionally altered in B.1.351 than in B.1.1.7.

Older adults make up the most vulnerable population to COVID-19 and therefore have been prioritized for vaccination[19]. We found similar age-dependent decline in FRNT50 titers against each lineage in our study (Fig. 1f–h and Table 1). These differences were highly significant for all three variants between subgroups of younger (20-50 y.o. $n = 25$) and older (>50 y.o. $n = 25$) adults in our cohort (Fig. 1i). There was no correlation between gender and neutralization titers after vaccination (Supplementary Fig. 3).

**Antibody response to natural SARS-CoV-2 infection.** In contrast to the spike-specific antibody repertoire raised by BNT162b2 vaccination, the antibody response to SARS-CoV-2 infection is more antigenically diverse[7]. Overall, RBD binding activities against all lineages were significantly lower in convalescent sera compared to vaccinee sera across all sample timepoints (1–301 days post-PCR positive) (Figs. 1a and 2a). Moreover, there was no observable difference in convalescent serum EC50 between RBD-WA1, RBD-B.1.1.7, and RBD-B.1.351 (Fig. 2a). In convalescent sera, there was also no clear correlation between variant-specific RBD binding and neutralization (Fig. 2c). To better capture the reduced antibody levels, we modified our ELISA protocol to reduce the limit of detection to 1:200 (compared with 1:1600 for vaccinee ELISAs). Differences in FRNT50 titer against WA1 and the VOCs were similarly reduced overall compared to vaccinee sera (WA1, GMTs 1:52.1 ± 4.3; B.1.1.7, 1:36.8 ± 3.0; B.1.351, 28.8 ± 2.3) but showed substantially less variability with a 1.8-fold drop for B.1.351 and a 1.4-fold drop for B.1.1.7 relative to WA1 (Fig. 2b and Supplementary Fig. 2). Many convalescent sera fell below the FRNT limit of detection (Fig. 2d, e): for WA1, 43% of convalescent cohort sera failed to neutralize ≥50% of input virus at the lowest dilution, and this proportion was even greater for the VOCs (B.1.1.7, 54%; B.1.351, 64%).

It remains unclear which factors, if any, are predictive of protection following recovery from COVID-19, however at least one study has shown a link between disease severity and final neutralizing antibody titer[4]. While our convalescent cohort did not show any significant correlation between FRNT50 and disease severity (Supplementary Fig. 4), this may be due to differences in

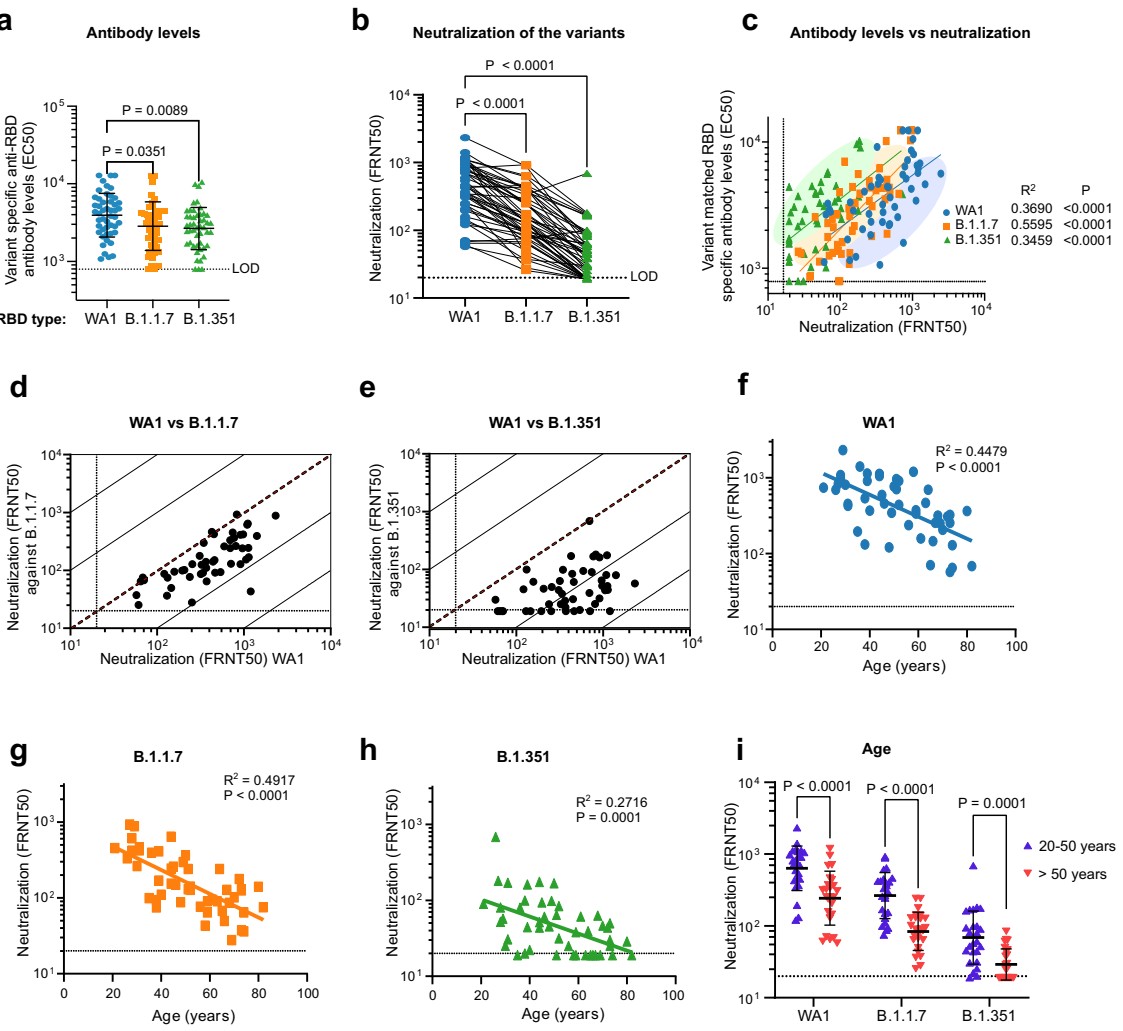

**Fig. 1 Serum antibody levels of BNT162b2 vaccine recipients and potency of sera to neutralize SARS-CoV-2 variants. a** Serum antibody levels (EC50) that recognize the spike RBD of the wild-type USA-WA1/2020 (WA1) (Blue circles), B.1.1.7 (Orange squares), and B.1.351 (Green triangles) variants are shown. The RBD-B.1.1.7 carries the N501Y mutation, the only RBD mutation present in the B.1.1.7 variant. The RBD-B.1.351 has the K417N, E484K, and N501Y mutations which are the only three RBD mutations present in the B.1.351 variant. $n = 51$ biologically independent samples. **b** Comparison of neutralization titers (FRNT50) between WA1, B.1.1.7 ($P = 0.0351$) and B.1.351 ($P = 0.0089$) for BNT162b2 vaccinee sera. $n = 50$ biologically independent samples. **c** Correlation of variant matched RBD-specific antibody levels and neutralization titers (FRNT50) of the WA1 virus and the two variants. **d**, **e** Correlations between neutralization titers of the B.1.1.7 (**d**) and B.1.351 (**e**) variants with the WA1 virus. The dotted diagonal lines indicate identical neutralization, and the solid diagonal black lines indicate 10-fold differences in neutralization. **f–h** Correlation between participant age and neutralization titer against WA1 (**f**) ($P < 0.0001$), B.1.1.7 (**g**) ($P < 0.0001$), and B.1.351 (**h**) ($P < 0.0001$). $n = 50$ biologically independent samples. **i** Effect of age range 20–50 years (blue triangle) and >50 years (red inverted triangle) on the neutralization potency among the BNT162b2 vaccine recipients (WA1,B.1.1.7 $P < 0.0001$, B.1.351 $P = 0.0001$). $n = 25$ biologically independent samples per age group. For **a**, **b**, **f–i**, data are presented as the mean ± SD of log transformed values; $P$ values are two-sided and include a Šidák multiple comparison correction. All experiments were performed in duplicate.

average disease severity between our cohorts. We additionally saw no correlation between neutralizing titer for any lineage with patient age, sex, or hospitalization for COVID-19 (Supplementary Fig. 4). Looking at the entire convalescent cohort, we see no significant correlation between FRNT50 and time between sample collection and first positive PCR test result for COVID-19 (Supplementary Fig. 4). However, subsetting the cohort into groups based on different ranges of days post PCR test, we see the median titer follow a similar trajectory to those reported in previous studies (Fig. 2f). Median titer values for all variants start (1–13 days) at a low point, increase to a maximum at 14–90 days, then decrease over 300 days. These results are in agreement with previous reports indicating that immunity resulting from vaccination peaks around 14 days after receiving the second boost, then wanes with a half-life of 69 days[21]. Although serum

neutralizing antibody titers apparently decrease over time, recent studies have shown that memory B cells can persist for at least a year following infection[22,23].

## Discussion

In this study we provide evidence of reduced antibody-mediated immunity to newly emerging SARS-CoV-2 variants B.1.1.7 and B.1.351 after immunization with the Pfizer-BioNTech COVID-19 vaccine or following natural infection. Our study involves a relatively large cohort, provides data well-balanced for gender and age distribution, controls for time since vaccination, and directly compares early-type and two newly emerging SARS-CoV-2 variants of global concern. Critically, we use authentic clinical isolates that display the native antigenic landscape of the virus, an

| Table 1 Demographic characteristics of study participants. | |
|---|---|
| **Convalescent serum donors** | |
| **Characteristic** | **Total (N = 50)** |
| Median age – year (range) | 56 (1–88) |
| Sex – no. (%) | |
| Female | 29 (58) |
| Male | 21 (42) |
| Symptomatic – no. (%) | |
| No | 4 (8) |
| Yes | 46 (92) |
| Hospitalized – no. (%) | |
| No | 34 (68) |
| Yes | 16 (32) |
| Admitted to ICU (subset of hospitalized) – no. (%) | |
| No | 12 (22.2) |
| Yes | 5 (9.25) |
| Median time between first positive COVID-19 PCR test and sample collection – days (range) | 188.5 (1–302) |
| **BNT162b2-vaccinated donors** | |
| **Characteristic** | **Total (N = 51)** |
| Median age – year (range) | 50 (21–82) |
| Sex – no. (%) | |
| Female | 28 (54.9) |
| Male | 23 (45.1) |
| Median time between vaccine doses – days (range) | 21 (20–22) |
| Median time between second dose and sample collection – days (range) | 14 (14–15) |

approach that provides the best possible examination of antibody activity against these viruses.

While it is likely that the resistance of some VOCs to neutralization is driven by accumulated mutations in the RBD and the rest of the spike protein, and there is evidence that high levels of RBD-binding antibodies is a meaningful correlate of protection from isogenic lineages[8,24], other features of host immunity may contribute to protection. Specifically, the neutralization titers seen in our convalescent subjects, while lower overall, have a smaller gap in neutralizing activity between WA1 and VOCs than in BNT162b2 vaccinees. This difference between convalescents and vaccinees suggests that SARS-CoV-2 infection may elicit more broadly cross-reactive and potentially cross-neutralizing antibodies, even with reduced affinity for mutant RBDs. This notion has a strong foundation in coronavirus research, as there is substantial cross-reactivity of anti-SARS-CoV spike antibodies with SARS-CoV-2 spike[25]. Indeed, risk of reinfection by VOCs may be driven by generally low serological responses in most COVID-19 patients, rather than the presence of RBD mutations that allow immune escape. Other arms of the adaptive immune response that we did not explore here, such as T cell immunity, could also contribute to cross-lineage immunity[26].

A particularly significant finding was the negative correlation between age and neutralizing antibody titer against VOCs in vaccinees, given that age is the predominant risk factor for severe COVID-19[27] and patients of advanced age stand to benefit the most from vaccination. Longitudinal studies of this and other cohorts could examine the durability of vaccine-induced immune responses and should be designed to resolve the nature of antibody responses induced by vaccination or natural infection that may correlate with broad cross-neutralization. At least one new study attempts answer this by estimating the minimum protective antibody titer using data from various published SARS-CoV-2 studies combined with validated influenza models[28]. At the same time, others have begun quantifying rates of vaccine breakthrough

infections for various VOCs[29]. Our data suggests that protection from natural infection-derived immunity wanes considerably by 6 months to a year post infection, and that vulnerability to the B.1.1.7 and B.1.351 viruses is likely higher than for the original SARS-CoV-2 lineage. Further work is needed to identify more precisely the protective antibody titer threshold, which will be particularly important for developing vaccines that will be effective in vulnerable populations, including those of advanced age, against future SARS-CoV-2 variants.

## Methods
This study was conducted in accordance with the Oregon Health & Science University Institutional Review Board (IRB#00022511 & #21230). Written informed consent was obtained for all study participants.

### Serum collection
*Vaccinated cohort - IRB#00022511.* Subjects were enrolled at Oregon Health & Science University immediately after receiving their first dose of the Pfizer-BioNTech COVID-19 vaccine. After written obtaining informed consent, 4–6 mL of whole blood were collected (BD Vacutainer® Plus Plastic Serum Tubes) and centrifuged for 10 min at 1000×g. A second blood sample was obtained 14–15 days after subjects received their second dose of the Pfizer-BioNTech COVID-19 vaccine. Samples were stored at −20 °C until sera were collected for neutralization assay. A subset of serological samples (n = 51) was randomly selected while maintaining equal gender representation, balanced age distribution, time between vaccination doses equal to 21 days ± 1 day, and time from boost to blood sampling equal to 14 days ± 1 day. Randomization was performed using R version 4.0.3 in RStudio version 1.2.5001.

*Natural infection cohort - IRB#21230.* Subjects with confirmed COVID-19 infection were part of a larger cohort of COVID-19 individuals at the Oregon Health & Science University. After obtaining written informed consent, 10 mL of whole blood was collected for serum (BD Vacutainer® Red Top Serum Tubes), and 40 mL of whole blood were collected for PBMCs and plasma (BD Vacutainer® Lavender Top EDTA Tubes). Serum tubes were centrifuged for 10 min at 1000×g. Samples were heat-inactivated for 30 min at 56 °C and stored at −20 °C until needed. A subset of serological samples (n = 50) from individuals with time post infection (determined by date of first positive PCR) ranging from 1 day – 10 months, a spectrum of disease severity scores and clinical disease ranging from asymptomatic to severe (hospitalized in the ICU) were chosen for this analysis.

**Cell culture**. Vero E6 monkey kidney epithelial cells (CRL-1586) were obtained from the ATCC. Unless otherwise stated, cells were maintained at all times in standard tissue culture-treated vessels in complete media (DMEM, 10% FBS, 1% nonessential amino acids, and 1% penicillin-streptomycin) at 37 °C and 5% $CO_2$.

**SARS-CoV-2 growth and titration**. SARS-CoV-2 isolates USA/CA_CDC_5574/2020 [lineage B.1.1.7] (NR-54011), hCoV-19/South Africa/KRISP-K005325/2020 [lineage B.1.351] (NR-54009), and USA-WA1/2020[30] [lineage A] (NR-52281) were obtained through BEI Resources and sequenced following a single passage (WA1/2020 isolate GenBank: SAMN18527778 and p1 GenBank: MZ344995; B.1.1.7 isolate GenBank: SAMN18527802 and p1 GenBank: MZ344998; B.1.351 isolate GenBank: SAMN18527801 and p1 GenBank: MZ344999). Sub-confluent monolayers of Vero E6 cells in 75 cm² flasks were inoculated with the p0 isolates and grown for 72 h, at which time significant cytopathic effect was observed for all strains. Culture supernatants were removed, centrifuged 10 min at 1000×g, and stored in aliquots at −80 °C. To determine titer, confluent monolayers of Vero E6 cells in 96-well plates were inoculated with tenfold serial dilutions of SARS-CoV-2 prepared in dilution media (Opti-MEM, 2% FBS) for 1 h at 37 °C, then covered with overlay media (Opti-MEM, 2% FBS, 1% methylcellulose) and cultured an additional 24 h. Overlay media was then removed, and plates were fixed for 1 h in 4% paraformaldehyde in PBS. To develop foci, cells were permeabilized for 30 min in perm buffer (0.1% BSA, 0.1% Saponin in PBS) and incubated with 1:5000 polyclonal anti-SARS-CoV-2 alpaca serum, generated by immunization of an alpaca with recombinant SARS-CoV-2 S and N proteins (Capralogics Inc.), for 2 h at room temperature. Plates were washed three times with wash buffer (0.01% Tween-20 in PBS), then incubated with 1:20,000 anti-alpaca-HRP (Novus #NB7242) for 2 h at room temperature. Plates were again washed three times with wash buffer and 30 μL of KPL TrueBlue substrate (Seracare #5510-0030) added to each well. Plates were incubated at room temperature for 20 min and imaged with a CTL Immunospot Analyzer, then foci were counted using CTL ImmunoSpot (7.0.26.0) Professional DC[25].

Additional SARS-CoV-2 isolates were propagated and titrated during the development of this assay. They included the three previously described clinical isolates: USA/CA_CDC_5574/2020 [lineage B.1.1.7] (NR-54011), hCoV-19/South Africa/KRISP-K005325/2020 [lineage B.1.351] (NR-54009), and USA-WA1/2020 [lineage A] (NR-52281) as well as two additional clinical isolates: hCoV-19/South

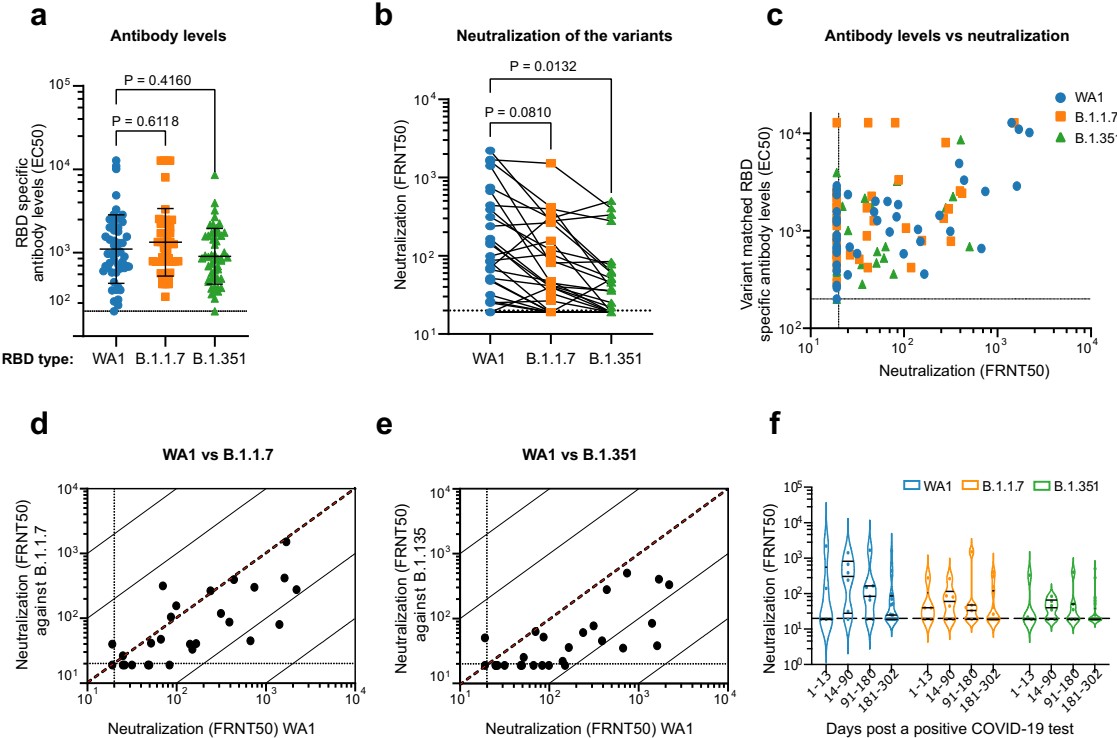

**Fig. 2 Neutralization of SARS-CoV-2 variants by convalescent serum. a** Quantification of serum antibody levels (EC50) that recognize RBD protein corresponding to the wild-type (WA1)(blue circles), B.1.1.7 (orange squares), and B.1.351 (green triangles) variants. $n = 50$ biologically independent samples. Data are presented as the mean ± SD of log transformed values. **b** Comparison of neutralization titers between WA1, B.1.1.7, and B.1.351 for convalescent sera. $n = 44$ biologically independent samples. **c** Relationship between convalescent antibody levels and neutralization (FRNT50) of the different viral variants. **d, e** Correlations between convalescent serum neutralization titer of the B.1.1.7 (**d**) and B.1.351 (**e**) variants with the WA1 virus. The dotted diagonal lines indicate identical neutralization, and the solid diagonal black lines indicate 10-fold differences in neutralization. **f** Violin plots indicating FRNT50 values for WA1 (blue), B.1.1.7 (orange), and B.1.351 (green), stratified by the number of days between the date of confirmatory COVID-19 PCR test and the date of serum sample collection. Black bars indicate median FRNT50 for each group. For **a, b, f** $P$ values are two-sided and include a Šidák multiple comparison correction. All experiments were performed in duplicate.

Africa/KRISP-EC-K005321/2020 [lineage B.1.351] (NR-54008) and hCoV-19/England/204820464/2020 (NR-54000). Substantial differences were noted in the focus phenotypes of these strains (Supplementary Fig. 5).

**SARS-CoV-2 sequencing**. Isolated viral RNA was subjected to first strand synthesis reverse transcription (RT) to produce single-stranded cDNA using Protoscript II (NEB), then amplified via pooled amplicon PCR using a 1,200 base pair overlapping amplicon strategy[31]. Individual sample PCR reactions were pooled, cleaned, and then subjected to shotgun sequencing library preparation[32,33]. Samples were then sequenced on a NextSeq500 using universal primers (Supplementary Table 2) with a custom sequencing protocol (Read 1: 50 imaged cycles; Index Read 1: 8 imaged cycles, 27 dark cycles, 10 imaged cycles; Index Read 2: 8 imaged cycles, 21 dark cycles, 10 imaged cycles; Read 2: 50 imaged cycles)[33]. FASTQ reads were quality trimmed and aligned to the reference genome (Gen-Bank: NC_045512), using BWA-mem. Quasispecies variant calling was performed using LoFreq[34]. Structural variant calling was conducted using Pindel[35]. Variant and coverage data were used to generate a per-sample consensus sequence, requiring a variant to be >50% of the total reads to be included. Consensus data used linage assignment using Pangolin[36]. Each cultured isolate consensus was compared to the original clinical isolate genome accessed from GISAID.

**SARS-CoV-2 FRNT**. Serial dilutions of patient sera and virus neutralization were carried out in duplicate, using separately prepared dilutions, in a 96-well plate format. Each sample was added in duplicate 1:10 to dilution media, and 4 four-fold serial dilutions were made spanning a range from 1:10 to 1:2560. An equal volume of dilution media containing 50 FFU of SARS-CoV-2 was added to each well (final dilutions of sera, 1:20–1:5120) and incubated 1 h at 37 °C. The virus-sera mixtures were then added to monolayers of Vero E6 in corresponding 96-well plates, incubated 1 h at 37 °C, and covered with overlay media. Fixation, foci development, and counting were carried out as described above in titration focus forming assay experiments. Focus counts were used to calculate percent neutralization by dividing by the average of positive control wells without patient serum treatment.

**Production of variant RBDs**. Site-directed mutagenesis was used to introduce mutations into Wild-type RBD (BEI resources #NR-52309). Purified SARS-CoV-2 WA1, B.1.1.7, and B.1.351 RBD protein was prepared by mammalian expression and Ni-NTA chromatography[25]. Sanger sequence confirmed (see Supplementary Table 2 for primer sequences) recombinant RBD lentivirus was produced using Lipofectamine 3000 (Invitrogen #L3000008) and used to generate stable HEK293F cells. Cells were allowed to grow for 3 days in Freestyle 293 expression media (Gibco # 12338018). Supernatant was collected by centrifugation at $1000 \times g$ for 10 min, then incubated with Ni-NTA beads for 1 h at room temperature. The beads were washed with ten column volumes of 20 mM imidazole in PBS, then eluted with 235 mM imidazole in PBS. The purified protein was buffer exchanged into plain PBS and concentrated by 10 kDa cutoff column and purity was assessed by $OD_{280}$ and SDS-PAGE.

**ELISA**. ELISAs were performed in biological duplicate 96-well plates (Nunc™ MaxiSorp™ #423501). Plates were coated 100 µL/well with purified wild-type SARS-CoV-2 RBD, RBD-501, or RBD-triple constructs at 1 µg/mL in PBS and incubated overnight at 4 C with rocking. Plates were then washed three times with wash buffer (0.05% Tween-20 in PBS) and blocked with 150 µL/well blocking buffer (5% nonfat dry milk powder and 0.05% Tween-20 in PBS) at RT for 1 h with rocking. Convalescent and vaccinated patient sera were initially diluted in Opti-MEM in the 96-well plate format used above. For the ELISA, diluted vaccinated and infected patient sera were further diluted in blocking buffer on the plate ($4 \times 4$-fold dilutions from 1:200 for infected patients; $4 \times 2$-fold dilutions from 1:1600 for vaccinated patients). After incubating at RT for 1 h with rocking, plates were washed three times. The secondary antibody Goat anti-Human IgG, IgM, and IgA (H + L) (Invitrogen, #A18847) was diluted in blocking buffer (1:10,000) and applied to the plates 100 µL/well. Plates were protected from light and incubated at RT for 1 h with rocking, then washed three times prior to the addition of the peroxidase activity detector 3,3′,5,5′-tetramethylbenzidine (TMB, Thermo Scientific Pierce 1-Step Ultra TMB ELISA Substrate #34029). The reaction was stopped after 5 min using an equivalent volume of 1 M $H_2SO_4$; optical density (OD) was measured at 450 nm using a CLARIOstar plate reader. $OD_{450}$ readings were normalized by subtracting the average of negative control wells and finally dividing by the average

maximum signal (95th percentile) for each unique coating protein in each experiment.

**FRNT50 and EC50 calculation**. Percent neutralization values for FRNT50 or normalized $OD_{450}$ values for $EC_{50}$ were compiled and analyzed using python (v3.7.6) with numpy (v1.18.1), scipy (v1.4.1), and pandas (v1.0.1) data analysis libraries. Data from biological replicates was combined and fit with a three-parameter logistic model. For FRNT50s, values were simultaneously calculated for individual biological replicates and patients for whom individual replicate FRNT50 values differed by more than 4-fold were excluded from further analysis. Final FRNT50 values below the limit of detection (1:20) were set to 1:19. Final $EC_{50}$ values below the limit of detection (1:1600 for vaccine cohort, 1:200 for natural infection cohort) were set to 1:1599 for the vaccine cohort and 1:199 for the natural infection cohort. EC50 and FRNT50 curves were plotted using python with the Matplotlib (v3.1.3) data visualization library.

**Statistical analysis**. Aggregated EC50 and FRNT50 values were analyzed in Graphpad Prism (v9.0.2). EC50 and FRNT50 data were log transformed and one-way ANOVA using the Šidák multiple comparison correction was used for columnated data while two-way ANOVA using the Šidák multiple comparison correction was used for grouped data. The reported statistical methods are indicated in the relevant figure legends. Comparison of fold reduction and 95% confidence intervals for EC50 and FRNT50 were generated using one-way ANOVA. Linear model fitting was performed on log transformed EC50 and FRNT50 data and statistical significance was determined by F test with a zero-slope null hypothesis. All P-values were two-tailed with a significance cutoff of 0.05. Patient samples with missing data points or demographic information were excluded from individual analyses which utilized those values.

**Reporting summary**. Further information on research design is available in the Nature Research Reporting Summary linked to this article.

## Data availability

The data associated with this study are provided in the supplementary materials or in public repositories. The data used to generate each figure are provided in (Source Data). The SARS-CoV-2 clinical isolate deep sequencing data generated in this study have been deposited in Genbank under accession codes found here: WA1/2020 consensus: MZ344995; B.1.1.7 consensus: MZ344998; B.1.351 consensus: MZ344999, raw sequence reads: PRJNA742050. The raw EC50 and FRNT50 curves are provided in (Supplementary Figs. 1 and 2). Raw focus counts and calculated FRNT50 and EC50 values are available on Zenodo at https://doi.org/10.5281/zenodo.5157016. The FRNT50 calculation code used in this study is available on Github at https://doi.org/10.5281/zenodo.5158655. Source data are provided with this paper.

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

## Acknowledgements

The authors thank the generous contribution of the many patients and vaccinees who participated in this study. In addition, we gratefully acknowledge the efforts of the entire OHSU COVID-19 serology study team. We also want to thank the following members of the Oregon SARS-CoV-2 Genome Sequencing Center at OHSU for their help in sequencing our samples: Sonia N. Acharya, Cierra N. LaBlanc, Kayla I. Carter, Sally Grindstaff, Brendan L. O'Connell, and Ruth V. Nichols. This study was funded in part by an unrestricted grant from the M.J. Murdock Charitable Trust, by NIH training grant

T32AI747225 on Interactions at the Microbe-Host Interface, and OHSU Innovative IDEA grant 1018784, and NIH R01AI145835.

## Author contributions

Concept and design: T.A.B., H.C.L., Z.L.L., M.E.C., W.B.M. and F.G.T.; acquisition, analysis, or interpretation of data: T.A.B., H.C.L., Z.L.L., S.K.M., F.J.C., J.B.W., J.R.G, Z.L., S.A.R.S., P.S., M.S., A.E.B., D.X.L., A.C.A., B.N.B., B.J.O., M.E.C., W.B.M. and F.G.T.; Drafting of the manuscript: T.A.B. and H.C.L.; critical revision of the manuscript for important intellectual content: T.A.B., H.C.L., Z.L.L., S.K.M., F.J.C., J.B.W., J.R.G, Z.L., S.A.R.S., P.S., M.S., A.E.B., D.X.L., A.C.A., B.N.B., B.J.O., M.E.C., W.B.M., and F.G.T.; statistical analysis: T.A.B., W.B.M. and A.E.B.; obtained funding: M.E.C., W.B.M. and F.G.T.

## Competing interests

The authors declare no competing interests.
