## [Peer Review File · Nature Communications]

REVIEWER COMMENTS

Reviewer #1 (Remarks to the Author):

Bates et al. report neutralization of SARS-CoV-2 variants by convalescent and vaccinated serum. The topic is important. The following concerns should be addressed.

Specific comments

1. Due to frequent mutations of variant after propagation, it is important to describe in the main text that no mutations were found in the variant viruses that had been used in the study.
2. Do the two mutants in Fig. 1A have all the RBD mutations from the B.1.1.7 and B.1.351 variants? If not, the authors should add data with complete RBD mutations from the variants. In this way, the ELISA data would be comparable to the rest of the virus neutralization results. Otherwise, the results are misleading in Fig. 1A & 1C.
3. In Fig. 1D & 1E, why was the correlation of FRNT50 titers between WA1 and B.1.1.7 much better than that between WA1 and B.1.351?
4. For Fig. 2F, what is the fold difference in FRNT50s between day 1 to day 301 post RT-PCT-positivity against three different viruses (WA1, B.1.1.7 and B.1.351)?
5. A previous study on the similar topic by Liu et al. (doi: 10.1056/NEJMc2102017) should be added to reference.

Reviewer #2 (Remarks to the Author):

To authors:

In this very timely work, Bates et al. tested the neutralization capacity of sera retrieved from a representative cohort of Biontech-vaccinated people or COVID-19 convalescent or infected patients. Using infectious SARS CoV-2 variants WA1, B.1.1.7 or B.1.351 in serum neutralization titration or ELISA assays with variant specific RBDs, the authors describe several important and surprising information:

1. The neutralization capacity of sera from vaccinated people is significantly weaker against the B.1.351 variant than against the WA1, with B.1.1.7 in the middle
2. The drop in neutralization capacity against the VoC SARS CoV-2 was more pronounced in vaccinees than in convalescent people.

3. The neutralization capacity of individual sera correlated only between Wa1 and B1.1.7 in vaccinees, but not between WA1 and B1.351 or in sera from convalescent patients in any constellation.

4. Variant specific ELISA strongly correlated with immune protection against the same strain, but only in sera from vaccinees. This correlation was weak to non-existent in sera from convalescent patients.

5. There was an age-dependent drop in neutralization capacity against all three variants, with numerous sera from older vaccinees showing no measurable neutralization against the B1.351 variant.

Overall these are important and new information that need to be communicated to the public very soon. It is important to note that two previously published studies that were performed to assess serum neutralization against novel VoC were done using pseudoviruses, while this is the first study to my knowledge relying on wild-type SARS CoV-2, rather than S expressing pseudoviruses. Therefore, the results showing a stronger reduction in neutralization capacity against VoC in the vaccinees than in the naturally infected people are important as they might indicate neutralization effects against VoC beyond the S protein.

I see no reason to substantially delay publication and request additional experiments, but I would like to request some minor modifications to improve readability and contextualize the data:

1. Two papers were published on serum neutralization capacity against the B1.1.7 VoC in BNT162b2 vaccinees, albeit using Pseudoviruses. (Muik et al. Science, Collier et al. Nature). These should be mentioned and discussed.

2. Some figure legends (1E, 2E show wrong labels on axes - B1.135 instead of 1.351)

3. The sentence in lines 98-99 is unclear and confusing as it stand and requires some rewriting.

4. The sentence in lines 126-128 is confusing and misleading. The authors mention severity of disease as a parameter for cross protection, but do not show a stratification in disease severity in the convalescent group, but rather on the time from diagnosis to serum collection.

5. In fact, this aspect remains lacking, should be available in the dataset and needs to be analyzed in more detail to avoid potentially misleading claims. In S4, the authors show that neither hospitalization (as bona fide indicator of severity) nor age predict neutralization potency. However, neither analysis is normalized on the other factor. It is reasonable to assume that older patients might have been hospitalized and that hospitalized patient might have developed higher antibody titers due to the intensity of the disease. Therefore, if the non-hospitalized fraction is substantially younger than the hospitalized one, the combination may result in effects that mask outcomes. The authors need to show the analyses for younger and older patients that were hospitalized (or not) and their neutralization capacities, in order to understand if trends observed in older vaccinees might be present in convalescent patients as well.

6. It would be important to provide some discussion about the relevance of the FRNT50 titer data for clinical outcomes. Are only people with FRNT50 below the detection threshold exposed to COVID? Is there an ELISA value that indicates a threshold for immune protection? The authors might wish to add, if available at all, references to neutralization values that were observed to be protective against infection and disease, in order to provide to the reader a more concrete measure of the value of their experimental assay for predicting clinical outcomes.

Reviewer #3 (Remarks to the Author):

Bates TA et al., examined the sensitivity of SARS-CoV-2 USA/WA1 (lineage A) and 2 variants (B.1.1.7 and B.1.351) to convalescent sera from COVID-19 patients and mRNA Pfizer vaccinated recipients. As recently reported in numerous articles, the authors find that B.1.351 is more resistant than the other strains to neutralization by sera from convalescent and vaccinated individuals. One advantage of the manuscript is the use of live virus, rather than pseudotypes, to test the neutralization activity of the sera. The authors analyzed 51 vaccinees and 44 patients. They report a negative correlation between neutralization titers and age. The paper is well-written and confirmatory of recent articles.

Specific points

1. The authors used an ELISA to detect antibodies against RDB-spike to detect antibodies against the different RBD spikes in vaccine recipients. Several sera are below the limit of detection in Fig. 1A. However, a neutralization is observed for all the participants, suggesting a lack of sensitivity of the ELISA. The ELISA in Fig. 1A should be performed as for convalescents individuals in Fig. 2A

2. It is surprising that only 43% of convalescent individuals neutralized WA1 SARS-CoV-2. Neutralizing antibody are known to persist several months after infection. The authors should further discuss this result. They could classify convalescent patients in different groups: 1-3 months, 4-6 months, 6-10-month post-infection.

Minor comments:

1. The figure 2B is not described in the text.

2. A set of recent publications on the neutralization activity of sera from convalescents and vaccinated individuals against variants (using pseudotypes or live virus) should be added as references and discussed (see recent issues of Cell, Nature and Nature Medicine). For instance Planas et al (Nature Medicine) recently used live virus to draw similar conclusions.

3. The authors mention in the abstract that they analyzed a “large” number of individuals. However, only 95 samples were analyzed.

Reviewer #4 (Remarks to the Authors):

In this manuscript, the authors examined the influence of the B.1.1.7 and B.1.351 variants on vaccine and infection induced antibody responses. The authors observed a reduction with both B.1.1.7 and B.1.351, with the latter variant showing greater reduction as compared to WT virus. Not surprisingly, age was negatively correlated with vaccine responses and RBD tracked with neutralizing antibody responses. Both of these latter findings have been described extensively in the literature. Most of these findings have been described in the literature and the findings are incremental.

Concerns:

1. For the COVID-19 individuals, details of the potential variant that these individuals were exposed to would be beneficial.

2. Was a spearman or pearson correlation analysis done in all of the plots? The R value is more informative of reporting than the R2 value.

3. The connecting lines between WA1, B.1.1.7 and B.1.351 are rather confusing. Why do some samples from the natural infection shown an increase in neutralization between B.1.1.7 and B.1.351?

4. Have the viruses been sequence verified? The viral stocks were propagated in VeroE6 cells, which will inevitably increase mutations within the furin site and polybasic site. This includes stocks provided by BEI Resources. Deep sequencing data should be provided on the working viral stocks. These mutations can have a profound impact on antibody neutralization.

Response to reviewers:

We thank the reviewers for their thoughtful and constructive comments and suggestions. We have addressed each of the reviewers' comments specifically, below. We believe that the revised manuscript is substantively improved, and we appreciate the important role the reviewers played in enhancing the quality of this study.

Reviewer #1 (Remarks to the Author):

Bates et al. report neutralization of SARS-CoV-2 variants by convalescent and vaccinated serum. The topic is important. The following concerns should be addressed.

Specific comments

1. Due to frequent mutations of variant after propagation, it is important to describe in the main text that no mutations were found in the variant viruses that had been used in the study.

We thank the review for this important suggestion. We collaborated with the Oregon SARS-CoV-2 Genome Sequencing Center at OHSU to perform deep sequencing of all our viral stocks and have included the results in our supplemental information. Moreover, we have uploaded all of these sequences in GenBank. For WA1, we noted one SNP (R682L) in our passage 1 sample compared to the clinical specimen, which is in the furin recognition site of the spike protein. This portion of the protein is unstructured, and we do not find evidence that this mutation impacts antibody epitope recognition or neutralization. For B.1.1.7 we noted no changes in the spike protein sequence in our samples from the original clinical isolate. For B.1.351 we noted a single SNP (Q677H) in a similar location to that of WA1, indicating that this region may be frequently altered during passage in VeroE6 cells. Full length consensus sequences and short read archives have been uploaded to GenBank. (Lines 220-223 and table S1).

We have expanded upon the current literature regarding the specific mutations we identified and their potential impact on neutralizing antibodies in the response to reviewer 4, question 4.

2. Do the two mutants in Fig. 1A have all the RBD mutations from the B.1.1.7 and B.1.351 variants? If not, the authors should add data with complete RBD mutations from the variants. In this way, the ELISA data would be comparable to the rest of the virus neutralization results. Otherwise, the results are misleading in Fig. 1A & 1C.

The RBD utilized by this study consisted of SARS-CoV-2 spike residues 319-541. Table S1 lists all amino acid changes present in the variants used by our study with those amino acids within the RBD highlighted in grey. Our variant RBD constructs do contain every amino acid change present in the clinical isolates used in the rest of the study. We agree that this is an important point to clarify, and we have updated the manuscript accordingly. (Lines 76-77, Figure 1 legend)

3. In Fig. 1D & 1E, why was the correlation of FRNT50 titers between WA1 and B.1.1.7 much better than that between WA1 and B.1.351?

We believe that the primary reason for the decrease in correlation seen in 1E compared to 1D is that the B.1.351 contains mutations which affect a greater proportion of commonly recognized epitopes and in turn a greater proportion of neutralizing antibodies. This is the primary driver of diminished

neutralization potency, and as many of the subjects' NT50 titers approach the limit of detection against B.1.351, observable correlations become weaker and, ultimately, disappear. We have expanded on this in the results section. (Lines 89-97)

4. For Fig. 2F, what is the fold difference in FRNT50s between day 1 to day 301 post RT-PCT-positivity against three different viruses (WA1, B.1.1.7 and B.1.351)?

To answer this question, we divided the cohort into 4 groups based on key milestones (14 days, 90 days, and 180 days). Using these groups, we find that individuals prior to 14 days post PCR+ show visibly reduced titers, which increase to a maximum from 14-90 days, then wane steadily from 91-180 and 181-302 days post PCR+. One caveat is that this is a cross-sectional cohort rather than a longitudinal cohort with repeat samples. Although the difference between the maximum (14-90) and minimum (181-302) groups was not significant, this is, likely due to the small sample size of the 14-90 (N=6). We used median titer for these groups instead of GMT because the distribution of the samples was not normal, and found large fold-changes (12.6 for WA1, 3.3 for B.1.1.7, and 2.1 for B.1.351). We generated violin plots of this analysis and replaced the previous figure 2F, which we moved to the supplement. We also cite new studies showing that while neutralizing antibody titers do seem to change over time, memory B cells do seem to persist for at least a year following infection. (Lines 132-140, Figure 2F)

5. A previous study on the similar topic by Liu et al. (doi: 10.1056/NEJMc2102017) should be added to reference.

We thank the reviewer for bringing this study to our attention. We have now included a reference to this study in our manuscript. (Line 38)

Reviewer #2 (Remarks to the Author):

To authors:

In this very timely work, Bates et al. tested the neutralization capacity of sera retrieved from a representative cohort of Biontech-vaccinated people or COVID-19 convalescent or infected patients. Using infectious SARS CoV-2 variants WA1, B.1.1.7 or B.1.351 in serum neutralization titration or ELISA assays with variant specific RBDs, the authors describe several important and surprising information:

1. The neutralization capacity of sera from vaccinated people is significantly weaker against the B.1.351 variant than against the WA1, with B.1.1.7 in the middle
2. The drop in neutralization capacity against the VoC SARS CoV-2 was more pronounced in vaccinees than in convalescent people.
3. The neutralization capacity of individual sera correlated only between Wa1 and B.1.1.7 in vaccinees, but not between WA1 and B.1.351 or in sera from convalescent patients in any constellation.
4. Variant specific ELISA strongly correlated with immune protection against the same strain, but only in sera from vaccinees. This correlation was weak to non-existent in sera from convalescent patients.
5. There was an age-dependent drop in neutralization capacity against all three variants, with numerous sera from older vaccinees showing no measurable neutralization against the B.1.351 variant.

Overall these are important and new information that need to be communicated to the public very

soon. It is important to note that two previously published studies that were performed to assess serum neutralization against novel VoC were done using pseudoviruses, while this is the first study to my knowledge relying on wild-type SARS CoV-2, rather than S expressing pseudoviruses. Therefore, the results showing a stronger reduction in neutralization capacity against VoC in the vaccinees than in the naturally infected people are important as they might indicate neutralization effects against VoC beyond the S protein.

I see no reason to substantially delay publication and request additional experiments, but I would like to request some minor modifications to improve readability and contextualize the data:

1. Two papers were published on serum neutralization capacity against the B1.1.7 VoC in BNT162b2 vaccinees, albeit using Pseudoviruses. (Muik et al. Science, Collier et al. Nature). These should be mentioned and discussed.

We thank the reviewer for bringing these studies to our attention. We have now included a reference to these studies in our manuscript. (Lines 72-74)

2. Some figure legends (1E, 2E show wrong labels on axes - B1.135 instead of 1.351)

We apologize for this oversight and all such errors have been corrected to our knowledge. (Figures 1,2 and their legends)

3. The sentence in lines 98-99 is unclear and confusing as it stand and requires some rewriting.

We have re-worded this sentence to indicate more clearly that the level of correlation between neutralization of WA1 compared to B.1.1.7 is greater than the correlation between WA1 and B.1.351. (Lines 89-97)

4. The sentence in lines 126-128 is confusing and misleading. The authors mention severity of disease as a parameter for cross protection, but do not show a stratification in disease severity in the convalescent group, but rather on the time from diagnosis to serum collection.

We agree with the reviewer, and we have re-evaluated this paragraph in its entirety due to improved data analysis suggested by multiple reviewers (Please refer to reviewer #1 question 4 for a more detailed description of how this has impacted the interpretation of this portion of our study). (Lines 123-140)

5. In fact, this aspect remains lacking, should be available in the dataset and needs to be analyzed in more detail to avoid potentially misleading claims. In S4, the authors show that neither hospitalization (as bona fide indicator of severity) nor age predict neutralization potency. However, neither analysis is normalized on the other factor. It is reasonable to assume that older patients might have been hospitalized and that hospitalized patient might have developed higher antibody titers due to the intensity of the disease. Therefore, if the non-hospitalized fraction is substantially younger than the hospitalized one, the combination may result in effects that mask outcomes. The authors need to show the analyses for younger and older patients that were hospitalized (or not) and their neutralization capacities, in order to understand if trends observed in older vaccinees might be present in convalescent patients as well.

We performed both univariate and multivariate analyses on the convalescent cohort data and found that age, disease severity, and days post PCR+ do not significantly predict WT FRNT50 levels, when controlling for other factors. The simultaneously large range of ages, disease severity scores, and days post PCR positive test among our convalescent cohort add great value to our study, but also limit the resolving power we have to look at all of these variables together.

6. It would be important to provide some discussion about the relevance of the FRNT50 titer data for clinical outcomes. Are only people with FRNT50 below the detection threshold exposed to COVID? Is there an ELISA value that indicates a threshold for immune protection? The authors might wish to add, if available at all, references to neutralization values that were observed to be protective against infection and disease, in order to provide to the reader a more concrete measure of the value of their experimental assay for predicting clinical outcomes.

Very little data exists that directly addresses the question of protective antibody threshold. One new study (Khoury et al. 2021) utilizes aggregated SARS-CoV-2 antibody titer and longevity data combined with models and protective threshold data from influenza to start making estimates. As the number of vaccinated individuals continues to rise, so too have anecdotal reports of vaccine breakthrough cases, however large-scale studies are still lacking. Another new study (Kustin et al. 2021), addresses breakthrough cases, including those with variants, within Israel. Israel was among the first to achieve high rates of immunization and thus will undoubtedly be the first to notice the signs of waning immunity and variant breakthrough. In this study, most cases are seen during the period before the vaccine reaches full efficacy, but some infections are occurring at later timepoints. However, the primary outcome of this study was the relative risk of the variants among vaccinated individuals compared to the wild-type virus, and no antibody neutralizing titer data was collected. This very important question remains essentially unanswered to this point. We have added elements of this discussion to the manuscript (Lines 172-180)

Reviewer #3 (Remarks to the Author):

Bates TA et al., examined the sensitivity of SARS-CoV-2 USA/WA1 (lineage A) and 2 variants (B.1.1.7 and B.1.351) to convalescent sera from COVID-19 patients and mRNA Pfizer vaccinated recipients. As recently reported in numerous articles, the authors find that B.1.351 is more resistant than the other strains to neutralization by sera from convalescent and vaccinated individuals. One advantage of the manuscript is the use of live virus, rather than pseudotypes, to test the neutralization activity of the sera. The authors analyzed 51 vaccinees and 44 patients. They report a negative correlation between neutralization titers and age. The paper is well-written and confirmatory of recent articles.

Specific points

1. The authors used an ELISA to detect antibodies against RDB-spike to detect antibodies against the different RBD spikes in vaccine recipients. Several sera are below the limit of detection in Fig. 1A. However, a neutralization is observed for all the participants, suggesting a lack of sensitivity of the ELISA. The ELISA in Fig. 1A should be performed as for convalescents individuals in Fig. 2A

We have re-evaluated our assay and have decided that our prior positive threshold was overly conservative and masked data that would have otherwise given useful insight. After lowering the threshold from 1:1600 (the lowest dilution used) to 1:800 (2-fold below the lowest dilution) we found that all of the data points became within the assay range. (Lines 79-80, figure 1A, 1C)

2. It is surprising that only 43% of convalescent individuals neutralized WA1 SARS-CoV-2. Neutralizing antibody are known to persist several months after infection. The authors should further discuss this result. They could classify convalescent patients in different groups: 1-3 months, 4-6 months, 6-10-month post-infection.

We have performed this analysis in response to this question and a similar one posed by reviewer 1 (please refer to reviewer 1 question 4 for a full writeup of our findings). We appreciate the reviewer's insight that this aspect of the data would yield interesting and important trends not seen during our initial analysis. (Lines 123-140)

Minor comments:

1. The figure 2B is not described in the text.

This has been remedied (Line 119)

2. A set of recent publications on the neutralization activity of sera from convalescents and vaccinated individuals against variants (using pseudotypes or live virus) should be added as references and discussed (see recent issues of Cell, Nature and Nature Medicine). For instance Planas et al (Nature Medicine) recently used live virus to draw similar conclusions.

Research on this topic is progressing at an extremely rapid pace, and there is now little doubt that both B.1.1.7 and B.1.351 are less sensitive to neutralization by early lineage SARS-CoV-2 immune serum. We have updated our discussion to include some of the findings which have been published after the submission of our manuscript in order to fit better into the current context and state of knowledge. (Lines 72-74)

3. The authors mention in the abstract that they analyzed a "large" number of individuals. However, only 95 samples were analyzed.

We use the word large in reference to similar studies available at the original time of submission. We agree that this language may be confusing in the context all SARS-CoV-2 studies which utilize higher throughput methods. We have updated our language to avoid such confusion. (Lines 21-32)

Reviewer #4 (Remarks to the Authors):

In this manuscript, the authors examined the influence of the B.1.1.7 and B.1.351 variants on vaccine and infection induced antibody responses. The authors observed a reduction with both B.1.1.7 and B.1.351, with the latter variant showing greater reduction as compared to WT virus. Not surprisingly, age was negatively correlated with vaccine responses and RBD tracked with neutralizing antibody responses. Both of these latter findings have been described extensively in the literature. Most of these findings have been described in the literature and the findings are incremental.

Concerns:

1. For the COVID-19 individuals, details of the potential variant that these individuals were exposed to would be beneficial.

We unfortunately do not have access to viral sequences for the convalescent participants.

2. Was a spearman or pearson correlation analysis done in all of the plots? The R value is more informative of reporting than the R2 value.

We did not perform separate correlation calculations, however the R^2 values of every linear fit is provided. The square root of the R^2 value is the pearson's correlation coefficient, where the slope of the line indicates whether the correlation is positive or negative. The residuals were all approximately normally distributed, but for good measure we also calculated spearman's rho for each of the linear fits presented. We have summarized the correlation coefficients below:

	Pearson's r	Spearman's Rho
Fig 1C:		
WA1	0.60745	0.69301
B.1.1.7	0.74800	0.77453
B.1.351	0.58813	0.66892
Fig 1F (WA1)	-0.66925	-0.65712
Fig 1G (B.1.1.7)	-0.70121	-0.68575
Fig 1H (B.1.351)	-0.52115	-0.54159

3. The connecting lines between WA1, B.1.1.7 and B.1.351 are rather confusing. Why do some samples from the natural infection shown an increase in neutralization between B.1.1.7 and B.1.351?

We do see a few outliers, particularly with the convalescent samples that appear to show increased neutralization of B.1.351 compared to B.1.1.7. We expected to see some heterogeneity in the data because these are patient samples. For the convalescent samples, we often cannot confirm if the initial infection was with a variant, or whether the participant suffered an additional infection between sample collections, which were up to 302 days apart. These uncertainties are why we feel it is important to have the sample sizes we used.

4. Have the viruses been sequence verified? The viral stocks were propagated in VeroE6 cells, which will inevitably increase mutations within the furin site and polybasic site. This includes stocks provided by BEI Resources. Deep sequencing data should be provided on the working viral stocks. These mutations can have a profound impact on antibody neutralization.

We thank the reviewer for their insightful feedback here. We have sequence verified our virus stocks using deep sequencing, which has now been included in the methods section. We are also including the deep sequencing results in this response, for your review:

(<https://www.ncbi.nlm.nih.gov/bioproject/PRJNA742050>).

In the WA1 viral stock, we found R682L which is within the polybasic furin cleavage site, and in the B.1.351 viral stock we found the Q677H mutation which lies immediately adjacent to the polybasic cleavage site. The Q677H mutation, in particular, has been found frequently in both Vero E6 propagated samples, such as those distributed by BEI resources, as well as in clinical samples, suggesting that this and other polybasic cleavage site mutations tends to arise both in culture and in the wild.^{1,2} Studies of neutralizing antibody escape mutations have done an excellent job of predicting emerging variant amino acid changes, but none have found polybasic cleavage site mutations to be impactful.³⁻⁶ Further, we found two studies which specifically tested the effect of the Q677H and other polybasic cleavage site mutations, both of which found no significant impact on neutralization by patient immune sera.^{7,8}

There are several different measurements which support this point, but we would call attention to Figure 2A of Tada et al. wherein the authors test convalescent patient sera for the ability to neutralize SARS-CoV-2 spike pseudovirus. In this plot, the authors compared D614G pseudovirus compared to COH.20G/677H which contains N501Y, D614G, and Q677H. As we can see, neither N501Y nor Q677H are sufficient to evade antibody binding compared to D614G alone. Klimstra et al. addressed whether these cleavage site mutations are commonly seen in live SARS-CoV-2 after propagation in Vero E6 cells and saw that they appear rapidly but are not believed to have any significant impact on antibody neutralization. Instead, cleavage site mutations reflect differences in cell surface proteases (between Vero E6 and human epithelial cells) that assist in viral entry, which occurs after the ACE2 attachment step that neutralizing antibodies typically block.

References:

1. Hodcroft, E. B. *et al.* Emergence in late 2020 of multiple lineages of SARS-CoV-2 Spike protein variants affecting amino acid position 677. *medRxiv* 2021.02.12.21251658 (2021) doi:10.1101/2021.02.12.21251658.
2. Sasaki, M. *et al.* SARS-CoV-2 variants with mutations at the S1/S2 cleavage site are generated in vitro during propagation in TMPRSS2-deficient cells. *PLoS Pathog* **17**, e1009233 (2021).
3. Starr, T. N. *et al.* Deep Mutational Scanning of SARS-CoV-2 Receptor Binding Domain Reveals Constraints on Folding and ACE2 Binding. *Cell* **182**, 1295-1310.e20 (2020).
4. Starr, T. N. *et al.* Prospective mapping of viral mutations that escape antibodies used to treat COVID-19. *Science* (2021) doi:10.1126/science.abf9302.
5. Greaney, A. J. *et al.* Complete Mapping of Mutations to the SARS-CoV-2 Spike Receptor-Binding Domain that Escape Antibody Recognition. *Cell Host Microbe* **29**, 44-57.e9 (2021).
6. Liu, Z. *et al.* Identification of SARS-CoV-2 spike mutations that attenuate monoclonal and serum antibody neutralization. *Cell Host & Microbe* **29**, 477-488.e4 (2021).
7. Tada, T. *et al.* Convalescent-Phase Sera and Vaccine-Elicited Antibodies Largely Maintain Neutralizing Titer against Global SARS-CoV-2 Variant Spikes. *mBio* **0**, e00696-21.
8. Klimstra, W. B. *et al.* SARS-CoV-2 growth, furin-cleavage-site adaptation and neutralization using serum from acutely infected hospitalized COVID-19 patients. *Journal of General Virology* **101**, 1156–1169.

REVIEWERS' COMMENTS

Reviewer #1 (Remarks to the Author):

The authors have appropriately addressed this reviewers' points.

Reviewer #2 (Remarks to the Author):

The authors have addressed my comments from the first round. Due to changes in WHO nomenclature, I advise them to alter the designations to Greek alphabet letters, but otherwise, I am fine with the outcome.

Reviewer #3 (Remarks to the Author):

The authors have addressed my concerns

REVIEWERS' COMMENTS

We thank all three reviewers for their valuable feedback and the role they played in improving the manuscript.

Reviewer #1 (Remarks to the Author):

The authors have appropriately addressed this reviewers' points.

We are glad to have satisfactorily addressed the reviewers comments.

Reviewer #2 (Remarks to the Author):

The authors have addressed my comments from the first round. Due to changes in WHO nomenclature, I advise them to alter the designations to Greek alphabet letters, but otherwise, I am fine with the outcome.

We have removed the language stating country of origin, and have replaced it with the new Greek letter designations. We prefer to keep the Pango codes for the majority of the manuscript because of their greater precision. We also added the Greek letter codes to the abstract to help orient readers (lines 27, 45-46)

Reviewer #3 (Remarks to the Author):

The authors have addressed my concerns

We are glad to have satisfactorily addressed the reviewers comments.